



# Impact of combined microphysical uncertainties on convective clouds and precipitation in ICON-D2-EPS forecasts during different synoptic control

Takumi Matsunobu[1], Amirmahdi Zarboo[2], Christian Barthlott[2], and Christian Keil[1]

[1]Meteorologisches Institut, Ludwig-Maximilians-Universität, Munich, Germany
[2]Institute of Meteorology and Climate Research (IMK-TRO), Department Troposphere Research, Karlsruhe Institute of Technology (KIT), Karlsruhe, Germany

**Correspondence:** Takumi Matsunobu (Takumi.Matsunobu@lmu.de)

**Abstract.** The relative impact of individual and combined uncertainties of cloud condensation nuclei (CCN) concentration and the shape parameter of the cloud drop size distribution (CDSD) in the presence of initial and boundary condition uncertainty (IBC) on convection forecasts is quantified using the operational convection-permitting model ICON-D2. We performed 180-member ensemble simulations for five real case studies representing different synoptic forcing situations over Germany and

inspect the precipitation variability on different spatial and temporal scales. During weak synoptic control, the relative impact of combined microphysical perturbations on area-averaged daily precipitation comprises about $\pm12\%$ which is around one-third the variability caused by operational IBC perturbations. The combined microphysical perturbations exceed the impact of individual CCN or CDSD perturbations. High CCN concentrations combined with a narrow CDSD show the largest decrease in precipitation. The combination of IBC and microphysical perturbations affect the extremes of daily spatially averaged rainfall

of individual members by extending the tails of the forecast distribution by 5% in weakly forced conditions. The responses are relatively insensitive in strong forcing situations. Visual inspection and objective analysis of the spatial variability of hourly rainfall rates reveal that IBC and microphysical perturbations alter the spatial variability of precipitation forecasts differently. Microphysical perturbations slightly shift convective cells but affect precipitation intensities while IBC perturbations scramble the location of convection during weak control. Cloud and rain water content is more sensitive to microphysical perturbations

than precipitation but slightly less dependent on the synoptic control. In contrast to the impact on precipitation, an increase in CCN concentration and shape parameter of CDSD has a significant positive impact on the formation of cloud water. Combined microphysical perturbations play a dominant role in cloud forecasts with a relative impact ranging between +79% and -62% on daily averaged vertically integrated cloud water, and between +57% and -35% on rain water content in weakly forced conditions. Thus microphysical uncertainty exhibits a relevant impact on cloud and rain water content and precipitation and its

impact largely depends on the prevailing synoptic control in mid-latitude warm-season weather forecasts.





# 1   Introduction

Weather forecasts are subject to various sources of uncertainty. The uncertainties originate from the chaotic nature of the atmospheric flow, the unknown true state of the atmosphere and an imperfect representation of physical processes governing atmospheric phenomena in numerical weather prediction (NWP) models. To account for the inherent uncertainties, ensemble prediction systems (EPS) are run that include perturbations to represent a range of known and unknown uncertainties throughout the forecasting window. Convective-scale EPS allow the determination and quantification of the relative importance of factors such as errors in the initial conditions, lateral boundary conditions and the model physics. Given the chaotic nature of convection it is often very difficult, not to say meaningless, to associate the sensitivity of convective precipitation forecasts with a certain model perturbation in a deterministic sense. An ensemble facilitates the distinction between systematic effects of perturbations and the chaotic signal.

One key element in regional EPS represents initial and lateral boundary conditions (IBC) perturbations, which are currently implemented at many national weather services by means of variational or ensemble data assimilation systems (Bannister, 2017). The regional EPS are driven by coarser (global) ensemble forecasts at the lateral boundaries. Another crucial component in ensemble NWP systems constitutes the formulation of the model error to consider the incomplete description of physical processes and to represent the subgrid-scale variability. Microphysical processes are essential to forming precipitation. Due to their inherent small spatial and temporal scale these processes are not only difficult to observe, but also to understand and to represent in NWP models. Moreover, many microphysical processes are insufficiently constrained by observations. The impact of parameter perturbations in microphysics parametrisation has been studied extensively with mostly deterministic ideal and real case experiments using a variety of NWP models and schemes. However, because of the large variability between schemes and cases, results from different systems are difficult to generalise. Investigations using a full convective-scale NWP ensemble system given IBC uncertainty are essential to address the relative impacts of microphysical uncertainty.

The impact of aerosols on microphysical processes in the formation of convective clouds and precipitation remains highly uncertain. The amount of aerosol in the atmosphere is one of the important factors influencing cloud formation. In general, more aerosol particles, which act as cloud condensation nuclei (CCN), activate condensation and increase the cloud water content while reducing the average size of cloud droplets. Smaller cloud droplet sizes and more narrow cloud droplet size distributions (CDSD) inhibit the generation and growth of raindrops primarily caused by the collision-coalescence process, thus prolonging the lifetime of clouds (Albrecht, 1989). A smaller droplet size shows a negative impact on precipitation in many cases, but the impact of CCN perturbations on precipitation is not always straightforward, as an increase in CCN provides more cloud water. Systematic responses of varied CCN concentration on precipitation are reported in numerous studies with a large variety depending on the used model and chosen case (Table 1 in Tao and Li, 2016). For example, Fan et al. (2009) shows negative impacts and its dependence on wind conditions using a bin microphysics scheme in idealised large-eddy simulations, while Wang (2005) and Baur et al. (2022) show positive ones attributed to convection enhancement and the suppression of rain evaporation, respectively, using two-moment bulk microphysics schemes with a grid spacing around 2 km. Keil et al. (2019) evaluate the impact of CCN uncertainties on precipitation and find that the spread of CCN-perturbed ensemble forecasts is





greater than the impact due to soil moisture. This effect is more pronounced under atmospheric conditions when the synoptic scale forcing is weak.

In current operational NWP systems grid-scale microphysical processes are mostly approximated by cost efficient one-moment bulk microphysics schemes due to the limitation of computational resources. In these parametrisations only the hydrometeor mass is prognostic. In two-moment microphysics schemes, that are currently mostly used in research, the number concentrations of hydrometeors can also be predicted. It is therefore possible to calculate mean particle radii at each model grid point and estimate more realistic CDSD. The shape of the CDSD is controlled by $\nu$, the pre-defined shape parameter. The width of the CDSD is not well constrained by observations and previous observational studies revealed a large range of the shape parameter between 0–14 (see e.g. Tab. 1 in Igel and van den Heever, 2017b). Thus the shape of the CDSD constitutes a potentially relevant source of microphysical uncertainty to be included in ensemble systems. In general, the broader the CDSD the more efficient the collision-coalescence process, since hydrometeor particles of various sizes are present in the atmosphere. Hence the shape parameter perturbation of the CDSD affects the cloud lifetime and raindrop growth as well. The importance of CDSD on precipitation forecasts has been evaluated by means of idealised simulations (e.g. Igel and van den Heever, 2017a). Recently, Barthlott et al. (2022) showed that narrowing of the CDSD can produce almost as large a variation in precipitation as a CCN increase from maritime to polluted conditions in realistic simulations.

The ultimate impact of various uncertainties described above varies greatly depending on the prevailing flow conditions. A successful approach to classify convective precipitation regimes is to focus on the strength and type of forcing that is driving convection. An objective measure for such a classification constitutes the convective adjustment time scale $\tau_c$ that provides a time scale over which CAPE (Convective Available Potential Energy) is consumed by precipitation. In strong synoptic forcing situations, when ascending motions caused by the synoptic scale flow lead to precipitation and the continuously produced CAPE is consumed immediately, the regime is in a kind of equilibrium, in which $\tau_c$ attains small values. On the other hand, in a weak synoptic forcing situation, CAPE accumulates until local phenomena that can initiate convection occur and precipitation shows an intermittent character. In this situation, $\tau_c$ can temporarily increase, especially before the initiation of convective precipitation in the afternoon. The strength of the synoptic control is found to influence the predictability and the impact of different types of perturbations on precipitation (Flack et al., 2016, 2018; Keil et al., 2019; Weyn and Durran, 2019).

The goal of the present study is to estimate the relative importance of certain microphysical uncertainties in view of the variability given by operational IBC conditional to synoptic control in central Europe. The microphysical perturbations comprise different CCN concentrations and shape parameters of CDSD. We conduct real case ensemble experiments for five days in August 2020 in different synoptic control situations using an operational NWP ensemble system. Specifically, the following research questions are addressed in this study:

- How large is the impact of individual and combined uncertainties on convective precipitation forecasts at different spatial and temporal scales?

- How weather regime dependent is that impact?





   – What is the impact on convective clouds and does the impact on cloud content translate into a comparable impact on
     precipitation?

## 2  Model and Experimental design

### 2.1  Model description

The numerical simulations are performed with the ICON-D2 (ICOsahedral Non-hydrostatic, version 2.6.2.2) model that covers
central Europe (see Fig. 2) and is operationally used at the Deutscher Wetterdienst (DWD) since February 2021. ICON-
D2 employs an icosahedral-triangular Arakawa-C grid with a grid spacing of 2 km (542040 grid points) and 65 vertically
discretised layers from the ground to 22 km above mean sea level. Its dynamical core is based on the non-hydrostatic equations
for fully compressible fluids as governing equations (see Zängl et al. (2015) for the details). Different from the operational
configuration, the two-moment bulk microphysics scheme (Seifert and Beheng, 2006) is used to investigate the impact of
number densities and size distributions of cloud water droplets. The ICON-D2 set-up is identical to Barthlott et al. (2022). Note
that the operationally used parameter perturbations in ICON-D2-EPS are turned off here to purely focus on the microphysical
perturbations representing the model error.

### 2.2  Perturbation design

To investigate the influence of uncertainties of CCN density and the shape of the CDSD, ICON-D2-EPS experiments with
180 members in total for each case, consisting of 20 different IBC, 3 different CCN concentrations, and 3 different shape
parameters of CDSD are performed (see experimental design in Fig. 1).

   The initial conditions are provided by pre-operational analyses produced by ICON-D2-KENDA (Kilometer-scale ENsemble
Data Assimilation (Schraff et al., 2016)). In August 2020 conventional measurements like radiosonde, aircraft, and ground-
based observations were assimilated in ICON-D2-KENDA using the Local Ensemble Transform Kalman Filter (LETKF; Hunt
et al., 2007). ICON-D2-KENDA produces 40-member ensemble analyses, while the first 20 analyses are used as initial condi-
tions for ICON-D2-EPS forecasts (as in operations at DWD) with 24 hour lead time due to limited computational resources.
Lateral boundary conditions are based on ensemble ICON global and EU-nest simulations initialised 3 hours before the initial
time of the ICON-D2-EPS experiments. The initial conditions for the global and EU-nest simulations are the operational anal-
yses provided by DWD with a grid spacing of 40 km for the global domain and 20 km for the nested EU domain. Different
from our ICON-D2-EPS simulations the one-moment microphysics scheme and the convection parametrisation for deep and
mid-level convection are active in the ICON global and EU-nest. The lateral boundary conditions are updated hourly using
data of the EU-nest forecasts at the lead times from 3 to 27 hours.

   In the Seifert and Beheng (2006) scheme, CCN activation rates are calculated using a lookup table of activation rates
empirically estimated by Segal and Khain (2006). To take insoluble CCN into account, certain portions of CCN are not activated
depending on their particle sizes (Seifert et al., 2012). Consistent with Barthlott et al. (2022) we vary CCN concentrations





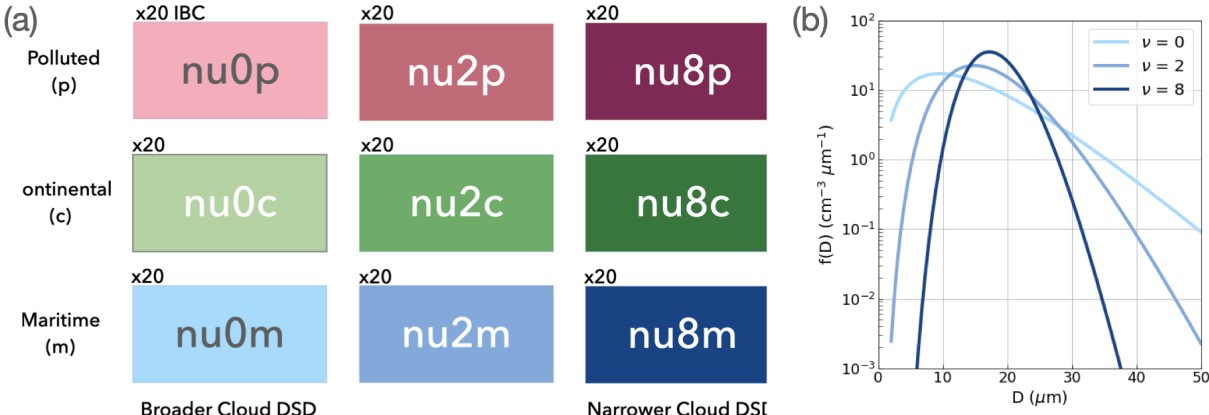

**Figure 1.** (a) Design of microphysically perturbed ensemble experiments. The colours used throughout the article indicate the nine different 20-member IBC sub-ensembles sharing the same combination of CCN and CDSD parameters. (b) Cloud drop size distribution with different shape parameter $\nu$ at fixed cloud water content ($QC = 1\,\mathrm{g\,m^{-3}}$) and cloud droplet number concentration ($QNC = 300\,\mathrm{cm^{-3}}$). $D$ denotes the diameter of the droplets.

between pristine conditions and extremely polluted conditions. We employ three CCN concentrations: maritime ($N_{CN} = 100\ \mathrm{cm^{-3}}$), continental ($N_{CN} = 1700\ \mathrm{cm^{-3}}$), and polluted ($N_{CN} = 3200\ \mathrm{cm^{-3}}$). The 'maritime' emulate clean, pristine conditions that have quite small numbers of CCN like over the sea. The 'continental' is the default setting that mimics the observed CCN concentrations for the European continental regions (Hande et al., 2016). The 'polluted' represents extremely polluted situations caused by, for example, massive wildfires and considerable anthropogenic emissions. Groups of ensemble members (called sub-ensembles) that share the same CCN concentration are named with suffixes m(aritime), c(ontinental) and p(olluted), as shown in Fig. 1a.

The size distribution of hydrometeors is approximated using the following generalised gamma distribution

$$f(x) = Ax^{\nu}\exp\left(-\lambda x^{\mu}\right) \tag{1}$$

where A is dependent on the number density of hydrometeor particles and $\lambda$ is a coefficient dependent on the average particle mass. The coefficients $\nu$ and $\mu$ are parameters that are pre-defined and fixed throughout a simulation. For example, with $\mu = \frac{1}{3}, \nu = -\frac{2}{3}$, we can obtain the so-called Marshall-Palmer distribution of raindrops. In this study we control the widths of the particle size distributions by varying the shape parameter $\nu$ (for details see Barthlott et al. (2022)). With increasing $\nu$ the CDSD becomes narrower and more skewed as shown in Fig. 1b, which means the number densities of particles close to the mean size increase. In this study $\nu$ is varied between 0, 2 and 8 to cover a wide spectrum of the possible shape parameter values (as in Wellmann et al. (2020); Barthlott et al. (2022); Baur et al. (2022)). Note that the default setting is the broadest CDSD $\nu = 0$.



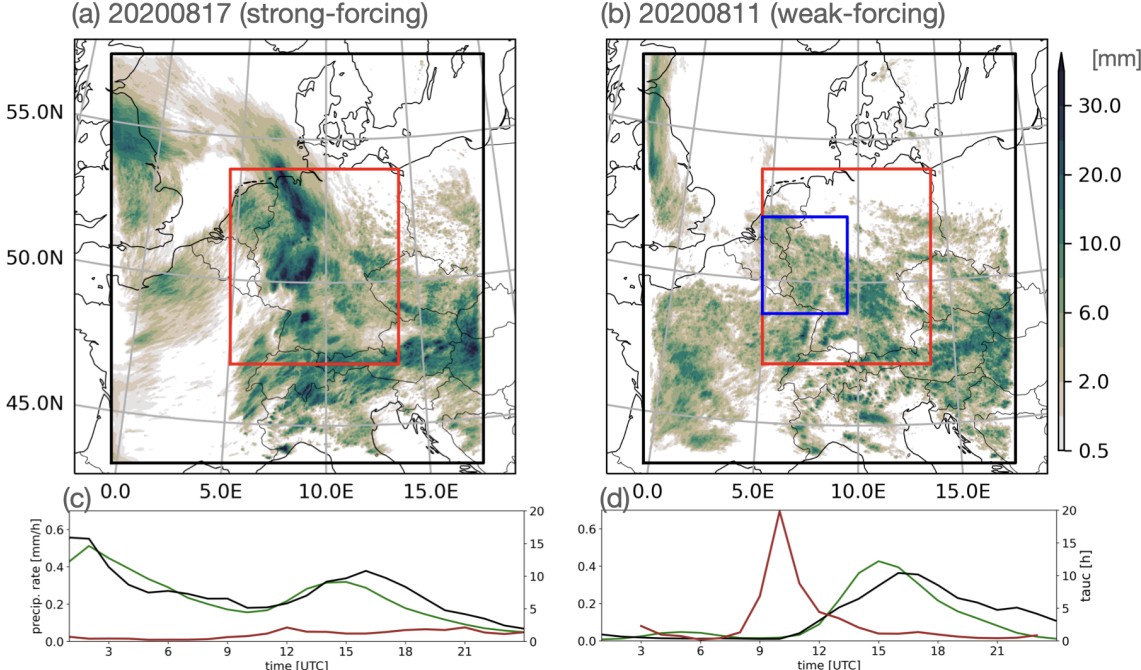

**Figure 2.** Daily accumulated precipitation on (a) a strongly forced day (17 August 2020) and (b) a weakly forced day (11 August 2020). Ensemble mean daily totals of the IBC sub-ensemble with nu0c microphysics are shown. The black rectangles indicate the ICON-D2 simulation domain, the red rectangles depict the German domain used for evaluation, and the blue rectangle depicts the central-western German domain used to inspect the spatial variability of rainfall patterns in Fig. 5. (c,d) The time series of area-averaged hourly sub-ensemble mean precipitation (green) and the convective adjustment time scale $\tau_c$ (red) complemented by the radar observed data (black) illustrate the different characteristics of both days.

## 3   Weather situation and case description

Two typical cases are selected for an in-depth investigation of the relative importance of the different uncertainties conditional to synoptic control. The 17 August 2020 represents a strong forcing situation associated with a weak low pressure system located over France that moved eastward towards Germany (not shown). The cyclonic flow favoured large-scale ascent initiating
convection, especially over the western part of Germany, resulting in widespread precipitation (Fig. 2a). There was rainfall from the start of the forecast, and the heaviest rainfall occurred at night followed by a gradual reduction of precipitation until noon (green in Fig. 2c). In the afternoon, there was a secondary peak of convective precipitation between 11 and 18 UTC. The daily maximum $\tau_c$ is less about 2 hours on 17 August 2020 (red line in Fig. 2a and Table 1). Such low values indicate that CAPE was immediately consumed by a continuous triggering of convection caused by synoptically forced ascending motion
characteristic in a so-called equilibrium regime.

On 11 August 2020 the precipitation texture shows a spotty distribution over southern Germany characteristic of convective precipitation in weak forcing situations (Fig. 2b). In a weak potential equivalent temperature gradient across central Europe





**Table 1.** List of case studies for which 180-member ICON-D2-EPS experiments were performed, indicating the date, the type of synoptic forcing, the daily maximum convective adjustment time-scale ($\tau_c$), and total precipitation (TP) of IBC sub-ensemble mean (TP$^{IBC}$) for default (nu0c) and the IBC sub-ensemble with maximum and minimum daily precipitation and its respective microphysical combination.

| Date | Forcing | $\tau_c$ [h] | Mean Total Precipitation TP$^{IBC}$ [mm/d] | | |
| --- | --- | --- | --- | --- | --- |
| | | | default | maximum | minimum |
| 11 August 2020 | weak | 20 | 2.67 | 2.95 (nu8m) | 2.42 (nu8p) |
| 12 August 2020 | weak | 7 | 1.58 | 1.73 (nu8m) | 1.45 (nu8p) |
| 13 August 2020 | strong | 3 | 3.72 | 3.90 (nu8m) | 3.60 (nu2p) |
| 17 August 2020 | strong | 2 | 5.72 | 6.00 (nu8m) | 5.51 (nu8p) |
| 18 August 2020 | weak | 6 | 3.79 | 4.07 (nu0m) | 3.51 (nu8p) |

(not shown) local trigger mechanisms (like convergence lines in the boundary caused by orography) initiate localised intense convection. The diurnal cycle nicely illustrates the typical development of convective precipitation starting with little precip-
itation in the morning and peak precipitation in the afternoon (green line in Fig. 2d). The daily maximum value of $\tau_c$ peaks at about 20 hours (red line in Fig. 2d), exceeding the 6 hour threshold used in previous work to distinguish different synoptic control in Europe (Keil et al., 2014, 2019; Kühnlein et al., 2014; Baur et al., 2018; Flack et al., 2018).

The comparison of the precipitation time series with area-averaged radar observations indicates the realism and fidelity of the ICON-D2-EPS forecasts (Fig. 2c,d). Characteristic values of the remaining three cases and their classification are presented
in Table 1.

## 4 Results

This section is structured in a scale-dependent manner. We start with broad scales inspecting area-averaged and 24 hour accumulated precipitation (total precipitation; TP) forecast of the 180-member ensemble for two cases. First we focus on the individual absolute amounts and their difference with respect to a sub-ensemble mean spanned by diverse IBC. This is followed
by relative differences stratified by the various uncertainties. The examination of the spatial variability rests upon finer space and time scales. The visual and objective investigation is based on the location of hourly rainfall rates of individual members. The discussion of the impact on cloud and rain water content is exemplified with area-averaged 24 hour mean values of the nine IBC sub-ensembles sharing the same microphysical setting for the weakly forced case. Finally statistics covering all five cases are presented.

### 4.1 Domain-averaged daily precipitation

The total precipitation of all individual 180 ensemble members is displayed in a scatter diagram for the prototype strong and weak forcing case. Fig. 3 shows the individual precipitation totals against the relative difference of any member to its respective

**Figure 3.** Scatterplot of total precipitation (TP) and relative difference of TP [%] regarding the combined microphysics sub-ensemble mean sharing the same initial and lateral boundary conditions (IBC) for the (a) strong and (b) weak forcing case. The nine coloured dots indicate the members belonging to the IBC sub-ensemble with the same microphysics configurations (as Fig. 1a) and connected dots with thin dashed lines indicate the members belonging to 20 microphysics sub-ensembles with identical IBC. The coloured lines show mean relative differences of IBC sub-ensembles. The dashed ellipse highlights member 17 discussed in the text.

microphysics sub-ensemble mean values sharing the same IBC. For instance, the relative difference of all nine members with combined microphysical perturbations but sharing the same IBC of member 17 range from -13% (nu8p experiment, TP=3.1

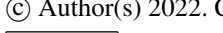



mm/d) to +11% (nu8m, TP=4.0 mm/d) given a sub-ensemble mean of $TP^{MP}$=3.6 mm/d (connected dots highlighted by dashed ellipse in Fig. 3b). At first sight and independent of the prevailing weather situation, the IBC perturbations largely control the precipitation amount. However, there is a systematic influence of microphysical perturbations on the daily totals. Ensemble members with low CCN concentration, that is clean conditions, show a positive impact on precipitation amount (blueish in Fig. 3). Increasing CCN concentration yields less and less precipitation (pinkish in Fig. 3).

During strong synoptic control the daily rainfall sums range from 4.6 mm/d to 6.9 mm/d for all 180 experiments (Fig. 3a). Looking at the extremes of the individual microphysics sub-ensembles (connected by dashed lines) reveals that these mostly comprise members with low or very high CCN contents (sub-ensembles nu8m (dark blue) and nu8p (dark pink)), but both having the narrowest CDSD, Fig. 1a) pointing towards the dominant influence of the CCN content among the combined microphysical perturbations. The mean impact of continental CCN concentrations to the sub-ensemble mean is on average

close to 0%, and those of maritime and polluted CCN concentrations show +3.5% and -3.5%, respectively. In contrast, the different shape parameters of CDSD show a non-systematic impact that is on average within ±2%.

    In the weak forcing situation, we find the same systematic responses to CCN concentrations, while average amplitudes of the microphysics' impact become larger than during strong control: increase of CCN from pristine (sub-ensemble nu8m) to very polluted conditions with the narrow CDSD (sub-ensemble nu8p) in Fig. 3) decreases relative precipitation from +11%

to -14% for the weak forcing case (+5% to -4% for the strong forcing case). Shape parameters of CDSD also exhibit a systematic impact in the weak forcing situation, whereas a CDSD's impact is hardly seen in the strong forcing situation. Narrower CDSD distributions give less precipitation, particularly during polluted conditions. The larger sensitivity to CDSD during weak synoptic control and a systematic decrease of precipitation with increasing shape parameter is consistent with Barthlott et al. (2022).

In both cases, close inspection of individual members sharing identical IBC shows a large uncertainty of the combined microphysics' impact (see e.g. light and medium green dots of highlighted member 17 in Fig. 3b) up to a reversal of the order of microphysical perturbed runs in terms of precipitation sums. Both aspects, the strong sensitivity to IBC (spread in IBC sub-ensemble, e.g. dark blue dots of sub-ensemble nu8m) in combination with the large variability within individual microphysics sub-ensembles illustrate the necessity to be cautious when interpreting results based on a deterministic approach

only to evaluate uncertainty.

    The overall response of domain and daily averaged precipitation sums to the various sources of uncertainty is summarised using boxplots in Fig. 4. Relative differences are calculated by subtracting a sub-ensemble mean sharing the same unperturbed parameters from each of the sub-ensemble members. For example, relative differences of full 180-member ensemble (black bars in Fig. 4) are calculated using all 180 members, those of the combined microphysics sub-ensemble (grey bars in Fig. 4)

are calculated using 9 members using identical IBC but different combinations of CCN and CDSD parameters.

    First, it becomes evident that the magnitude of the impact of the various uncertainties largely depends on the synoptic control. The 180-member ensemble including IBC and microphysical uncertainty shows the largest variability during weak control in agreement with previous studies (Barthlott and Hoose, 2018; Schneider et al., 2019; Keil et al., 2019). The extremes in daily precipitation of individual members deviate from the ensemble mean by +50% to -40%, with an interquartile range

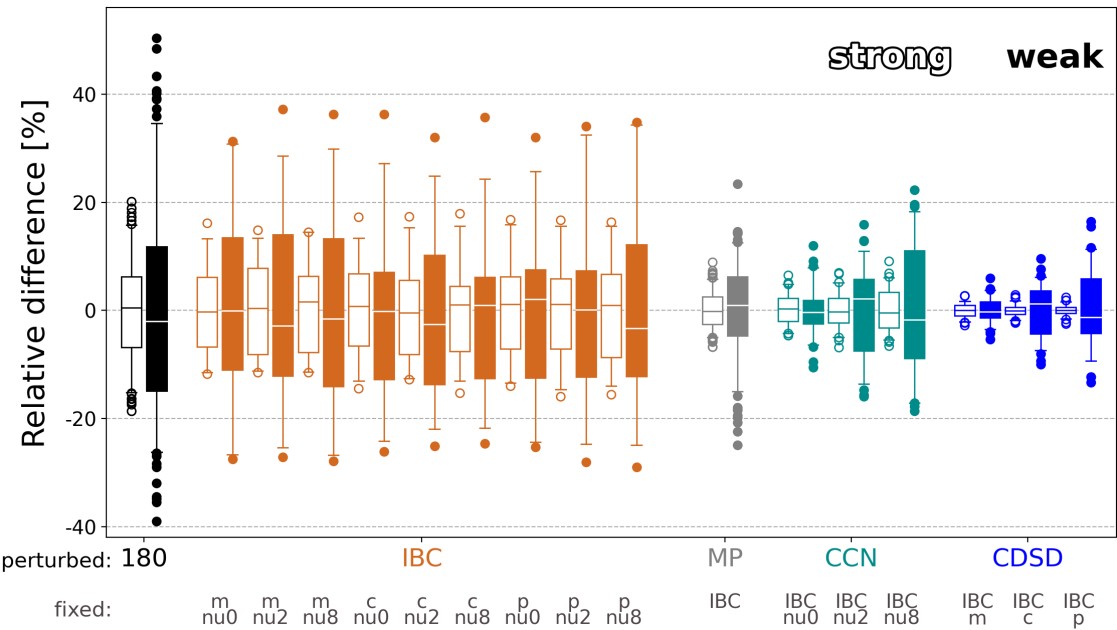

**Figure 4.** Statistics of relative differences of total precipitation (TP) of the members belonging to various (sub-)ensembles. The perturbations (upper x labels in colour) and different fixed configurations (x labels below) are indicated. The bars, boxes, whiskers and dots show medians, interquartile ranges, 5th and 95th percentiles and outliers, respectively. Open boxes represent strong synoptic control (17 August), filled boxes weak control (11 August). 180 (MP) is the abbreviation of the full (combined microphysics) ensemble.

of ±15%. The IBC sub-ensembles show a maximum range of +38% to -30% in daily sums during the weak forcing situation (filled orange dots of IBC in Fig. 4). Although their medians and interquartile ranges have some variability among the different microphysics configurations, no systematic dependence is found and the variability between the 9 IBC sub-ensembles is statistically insignificant. A corresponding behaviour is found for the strong forcing case with smaller amplitudes between +15% and -12% (open orange dots in Fig. 4).

Secondly, the combined microphysical perturbations (grey dots and boxes in Fig. 4) show a maximum relative impact of +22% to -25% for the weak forcing case, and ±10% for the strong forcing case. Interestingly, the impact of individual CCN perturbations show a clear dependence on the CDSD shape, and vice versa. CCN's impact is smallest (±10%) with a broad distribution (shape parameter $\nu = 0$), and increases to +22% to -20% with narrower distributions (increase of shape parameter). The impact of CDSD perturbations also increases with an increase of CCN concentration. This steady increase of impact is also 215 found in the CCN concentrations during strong forcing, while the shape of CDSD shows a small sensitivity only. Precipitation reacts more sensitive to microphysical perturbations during weak control.

In summary, IBC uncertainties dominate the impact on total precipitation, while the combined microphysical uncertainties play a secondary role. CCN has a larger impact than CDSD. Collective perturbations of CCN and CDSD enhance each other and show larger extremes in rainfall totals than individual CCN and CDSD perturbations. However, the interquartile range becomes





smaller than those of the CCN sub-ensembles with fixed shape parameters ($\nu = 2$ and 8) corresponding to a narrower CDSD. While the interquartile range of the 180-member ensemble and the individual IBC sub-ensembles is similar (between +10% and -15%), the extremes in the 180-member ensemble surpass the IBC variability by +15% and -10%. Thus, the combination of IBC and microphysical uncertainty affects the magnitude of the extremes while keeping the interquartile range fairly unaffected.

### 4.2   Spatial variability based on hourly rainrates

To address the question of how IBC and microphysical uncertainties affect convective precipitation on different spatiotemporal scales we now move from area averages to the kilometre scale and from daily to hourly accumulations. The fractions skill score (FSS; Roberts and Lean, 2008) and its variant believable scale (Dey et al., 2014; Bachmann et al., 2020) are used to objectively assess differences in spatial variability caused by different sources of uncertainty. But first we apply subjective visual inspection on selected precipitation fields to illustrate differences.

In Fig. 5 a snapshot of hourly precipitation over central western Germany at 16 UTC for the weak forcing case (11 August) exemplifies the different impact of IBC and microphysical perturbations. This day is chosen because of the stronger impact of the perturbations during weak synoptic control, 16 UTC represents the time of maximum afternoon precipitation within the diurnal cycle of convective precipitation (see Fig. 2b), and the displayed subdomain clearly depicts the typical popcorn-type precipitation structure. In Fig. 5 the transient character of individual cells is juxtaposed for four different experiments: three of

them share the identical IBC, CCN concentration and shape parameters of CDSD, respectively.

At first glance, it becomes evident that the microphysical perturbations result in a similar rainfall distribution (Fig. 5a, b, c), whereas the member driven with different IBC shows a considerably different rainfall field (Fig. 5d). The direct comparison of the location of intense precipitation caused by the different perturbations relative to the 99th percentile of simulation nu8p (black contours in Fig. 5) shows that convective cells of simulations nu0p (broad CDSD, polluted) and nu8m (narrow CDSD, maritime) are either at the same location or in the vicinity. Some weak rain cells (like in the southeast of Luxemburg in Fig. 5a)

are intensified by decreasing CCN and shape parameters of CDSD, thus in agreement with the spatiotemporal integrated rainfall signal. Positions of strong rain cells are shifted by the CCN perturbation at a scale of 20-30 kilometres, whereas an increase of the shape parameter of CDSD hardly shows a clear difference. The relatively small impact of CDSD perturbations in maritime CCN conditions is consistent with earlier findings discussed in Fig. 3 and Fig. 4. The visual inspection of many scenes of hourly rainfall caused by convective cells confirms the systematic behaviour of microphysical perturbations with

stronger precipitation in low CCN concentration and broad CDSD conditions (not shown).

To briefly summarise the visual inspection, we can state, that in a clean CCN environment, CDSD perturbations do not significantly affect the location of strong precipitation, whereas CCN perturbations shift the location by a few tens of kilometres. However, in polluted CCN conditions, both CCN and CDSD perturbations have an impact on spatial variability at almost the same scale. While microphysical perturbations keep the general spatial structure, IBC perturbations largely alter the position of

convective cells. Thus microphysical perturbations primarily impact the precipitation amount by changing their precipitation intensity rather than by feedback on dynamical fields and triggering new cells. Visual inspection of rainfall patterns of the



**Figure 5.** Snapshot of hourly precipitation at 16 UTC for the weak forcing case (11 August). Member 2 of IBC sub-ensembles (a) nu8p, (b) nu0p, (c) nu8m and (d) member 1 of nu8p in the central western part of Germany (see Fig. 2). Black contours indicate grid points that have a larger value than the 99th percentile value in the nu8p sub-ensemble of member 2.

strong forcing case results in similar findings: minor shifts of rain cells in microphysics sub-ensembles and a smaller impact of CDSD perturbations (not shown).



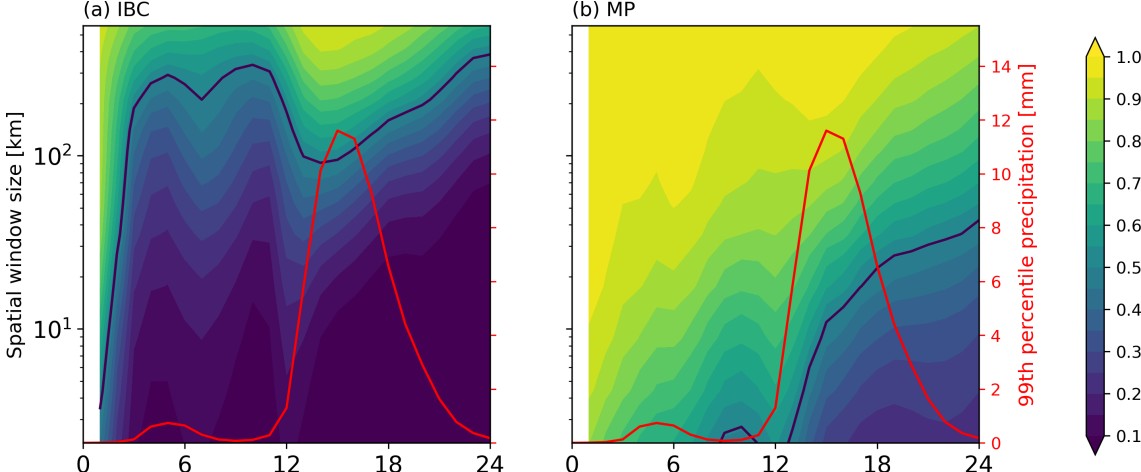

**Figure 6.** Ensemble mean FSS values of hourly precipitation calculated across scales ranging from 2 to 560 km across the German domain for the weak forcing case 11 August. The IBC sub-ensembles' mean FSS is depicted in panel (a), and the combined microphysics sub-ensembles' mean FSS in panel (b). The black lines show believable scales of mean FSS. The red lines (right axis) show the time series of mean 99th percentile value of hourly precipitation.

To quantify the spatial (dis-)agreement of hourly precipitation fields in the various simulations we employ the FSS, a spatial score that shows the similarity between two binary fields within a predefined neighbourhood scale. If the number of grid points with value of 1 within a certain neighbourhood of each grid point is equal between two fields, the FSS is 1.0, which means the compared two fields are identical. FSS becomes small as the difference between two fields gets larger, and it becomes 0.0 when only one of the fields has values and the other has a complete miss in the respective neighbourhood. In this study, we use the

99th percentile of hourly precipitation as the threshold to generate a binary field to take into account the strong diurnal cycle of rainfall intensity and to keep the number of grid points used for FSS calculation constant, and the 99th percentile seems a good threshold to well capture positions of convective cores (see contours in Fig. 5). The neighbourhood size is varied from 2.2 km (1 grid point) to 563.2 km (256 grid points) and FSS is calculated over Germany. Since FSS is a score calculated between two fields, we need to carefully consider how to compute an ensemble FSS. Following Dey et al. (2014), we calculate the FSS

for all combinations of ensemble members belonging to a sub-ensemble. For instance, FSSs for an IBC sub-ensemble (with 20 different IBC) can be calculated 20 * 19 / 2 = 190 times. Since there are 9 IBC sub-ensembles in this study, the number of overall FSSs that shows the impact of IBC perturbations is 190 * 9 = 1710. Accordingly, the numbers of FSSs for combined microphysics, CCN, and CDSD sub-ensembles are 7200, 180, and 180, respectively. Mean values of the FSSs are shown in Figs. 6 and 7 to objectively represent the spatial variability given by various kinds of uncertainties.

In addition, we use the believable scale (Dey et al., 2014; Bachmann et al., 2020) to characterise a typical length scale that estimates the spatial difference between two fields. The believable scale is defined as the neighbourhood size when the FSS exceeds a threshold defined by $FSS \geq 0.5 + \frac{f_0}{2}$ where $f_0$ is the fraction of grid points considered in the FSS calculation (the 99th percentile threshold gives $f_0 = 0.01$). Since the FSS is applied on precipitation fields above the 99th percentile values,



the believable scale can be considered in this study as a scale showing how large a mismatch of intense convective cells is.
Note that there is a difference between the believable scale of a 'mean FSS' (e.g. black line in Fig. 6) that represents a scale of
(dis-)agreement given, say, an ensemble mean FSS value and the mean over many believable scale values of paired member-
to-member comparisons (Fig. 8). The ensemble mean FSS is useful for an intercomparison of the average impact given by
different perturbations in general, whereas the mean of member-to-member believable scales (Fig. 8) provide a scale of actual
(dis-)agreement of certain scenes, for example, the precipitation patterns shown in Fig. 5.

Time-space diagrams of the ensemble mean FSSs given by (a) IBC and (b) combined microphysical perturbations for the
weak forcing case are depicted in Fig. 6. Low FSS values represent large spatial deviations between the location of intense
convection, hence a larger spatial variability. The variability due to the IBC perturbations is considerably larger than the one
forced by combined microphysical perturbations. However, and typical for days under weak control, convective precipitation
only forms in the late morning (see e.g. time series in Fig. 2b and red line depicting the 99th percentile of hourly precipitation
in Fig. 6). The value of 99th percentile of hourly precipitation amounts to 1 mm/h only at 12 UTC and precipitation is mostly
negligible before. Interestingly, at the onset of convective precipitation at 12 UTC the believable scale exhibits a dip and
the spatial variability decreases to slightly less than 100 km and thereafter continuously increases throughout the convective
period until the evening. The reduction of the variability represents that location of convective precipitation in the afternoon
is constrained by steady, non-perturbed factors forcing the dynamical fields involved in cloud and precipitation formation like
orography. After 22 UTC the hourly precipitation rates amount to less than 1 mm/h and the corresponding believable scale
exceeds 200 km as before the onset of convection in the night and morning. In contrast, the spatial disagreement caused
by combined microphysical perturbations is smaller and the mean believable scale amounts to only 16 km at the peak of
precipitation at 16 UTC (Fig. 6b). Apparently, the impact of microphysical perturbations on precipitation acting on many
pathways needs time and starts at a much lower spatial scale than IBC perturbations.

At first sight, individual perturbations of CCN and CDSD show a similar growth of FSS as the combined microphysical
perturbations (Fig. 6b and Fig. 7). Close inspection reveals, that the believable scale of CCN perturbations (black line in
Fig. 7b) starts to increase at the onset of the precipitation, 12 UTC, one hour before CDSD perturbations (Fig. 7a). The CDSD
believable scale grows more slowly and is always smaller (roughly 50%) than that of combined microphysical perturbations.
Since changes in CCN have a direct influence on the cloud condensation process, while the shape parameter of CDSD affects
ensuing microphysical processes, this time shift is plausible. Interestingly, the CCN perturbed believable scale reaches 40 km
after 22 hours, the same length scale as the believable scale of the combined microphysical perturbations. In contrast to the
impact on precipitation amount, combining two kinds of microphysical perturbations does not increase the spatial variability.

The uncertainty of CCN concentrations has a larger impact than shape parameters of CDSD on the spatial variability of
intense precipitation cells. Now we can ask if this behaviour is by chance and if this finding holds for other thresholds or per-
centiles, respectively. For this reason we performed additional white noise (WNoise) ensemble simulations with 20 different
IBC but only for the 'default' nu0c configuration to examine whether the spatial variability caused, for instance, by microphys-
ical perturbations differs from the impact of random, tiny differences in the temperature field. Following the method of Selz
and Craig (2015) the virtual potential temperature field is perturbed by a non-biased Gaussian noise with a standard deviation



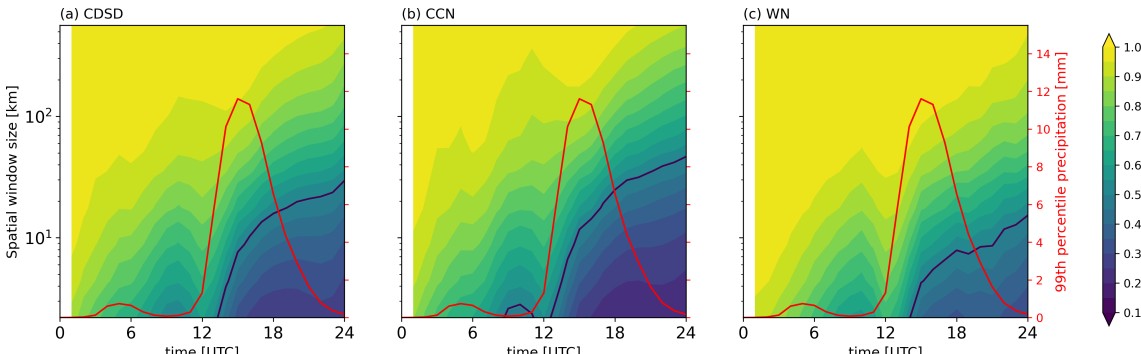

**Figure 7.** As Fig. 6, but for the (a) CDSD, (b) CCN and (c) WNoise sub-ensembles.

of 0.01 K at all grid points of the entire model atmosphere at initial time. The comparison of the microphysically perturbed
ensemble with a pure white noise (WNoise) experiment shows a similar onset and increase of spatial variability (Fig. 7c). The
spatial variability caused by CCN and CDSD perturbations is, however, larger than the effect of the WNoise perturbations. At
16 UTC, the mean FSS of WNoise simulations is close to 1 at scales larger than 80 km, and the believable scale is about 5 km.
Thus the effect of microphysical uncertainty on the spatial precipitation fields is systematically exceeding the effect of tiny er-
rors at initial time in the WNoise experiment. Less intense precipitation cells detected by the 95th percentile threshold indicate
a similar albeit slightly smaller variability due to IBC and microphysical perturbations (not shown). Using a 90th percentile
threshold on hourly precipitation results in values lower than 0.1 mm at all forecast hours and gives no extra information.

To further elucidate the combined microphysical perturbations and the interdependence of one perturbation (say CCN)
when the other (CDSD) is kept constant in the presence of IBC uncertainty, time series of all believable scales calculated
between every combination of ensemble members are illustrated in Fig. 8. The bold lines in Fig. 8a clearly reveal that CDSD
perturbations result in a spatial variability at different length scales depending on a certain fixed CCN concentration during
weak synoptic control. In clean air conditions (maritime, dark blue lines in Fig. 8a), the mean believable scale attains 10 km
roughly 3 hours after the onset of the believable scale's growth. At 22 UTC, towards the end of the diurnal cycle, the value
increases to 15 km. On the other hand, for polluted conditions (dark red and green lines), the mean believable scales reach larger
values, 15 km at 16 UTC and 30 to 40 km at 22 UTC. The mean length scale of disagreement given by the CDSD perturbations
in polluted conditions (high CCN concentrations) is twice as large as in clean conditions (low CCN concentrations). Note,
however, that there is big variability among the pairs of ensemble members, hence the IBC dependence is larger than the
impact of the background CCN condition. A similar systematic dependence can be found for the CCN perturbations' impact
with different fixed CDSD shape parameters. The mean believable scale with the broadest CDSD (lightest grey lines in Fig. 8b)
reaches 10 km at 16 UTC and 50 km after 22 hours lead time. With the narrowest CDSD (black lines), the mean believable
scale of CCN perturbations is 20 km at 16 UTC and increases to 100 km later. Interestingly, the mean believable scale with
the narrowest CDSD is by a factor of 2 larger than the broadest CDSD. This relationship is similar to that found in spatially
averaged precipitation amounts, namely polluted CCN and narrow CDSD conditions lead to larger variability (Fig. 4).

**Weather and Climate Dynamics Discussions**

**Figure 8.** Time series of FSS believable scales of hourly precipitation for every combination of (a) the CDSD and (b) CCN sub-ensemble for the weak forcing case across the German domain. In (a) dark pink, dark green and dark blue lines indicate simulations with polluted, continental and maritime CCN condition, respectively. In (b) black, dark grey and light grey lines indicate scales with the narrow, intermediate and broad CDSD. Bold lines with circles indicate mean values of FSS believable scales sharing the same perturbation. The red lines (right axis) show time series of mean 99th percentile value of hourly precipitation. Panels (c) and (d) show the results for the strong synoptic-forcing case.

In strong synoptic control, the situation is slightly different (Fig. 8c,d). The believable scales only start to grow from 7 UTC onwards, and the mean values finally reach a neighbourhood size of 30 km at 22 hours lead time. This monotonic pattern of the perturbation growth is the same as the weak forcing case. However, the mean believable scale for clean CCN conditions is





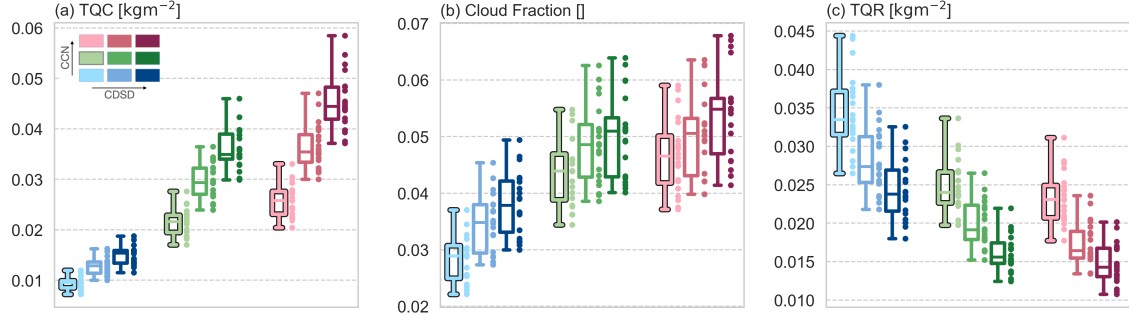

**Figure 9.** Box and swarm plots for 24h-mean (a) domain-averaged total column cloud water content, (b) cloud fraction, and (c) domain-averaged total column cloud water content over Germany for the weak forcing case. The boxplots and dots illustrate the same data set, but the dots represent individual IBC sub-ensemble members. The colours are based on the combination of microphysical configurations shown in Fig. 1. Boxplots show medians, interquartile ranges, maximum and minimum values.

larger than for the weak forcing case at 22 UTC (dark blue bold lines in Fig. 8a and c). There is no systematic difference in the mean believable scale caused by CDSD perturbations in the presence of various, yet fixed CCN concentrations (Fig. 8c). On the other hand, given narrower CDSD, the CCN perturbations cause a slightly larger spatial variability (Fig. 8d). Nevertheless, a difference between the broadest and narrowest CDSD is less pronounced in comparison to the weak forcing case (10-15

km difference in strong control versus 30 km in weak control at 22 UTC). It is interesting to note that the impact of the microphysical perturbations on the spatial precipitation pattern only starts to appear in FSS after 7 hours lead time, although there is continuous rainfall since forecast initialization during the strong forcing case. Thus microphysical perturbations need a much longer spin-up time than IBC perturbations to modulate dynamical fields eventually resulting in precipitation at different locations (see Fig. 8c,d).

## 4.3 Impact on cloud and rain water content


We now inspect how the various uncertainties impact the cloud and rain water content, both being important precursors in the complex process chain to form precipitation. Since we find similar systematic responses in both weather situations, we show results for the weakly forced case only.

Distributions of vertically integrated cloud water content (TQC) averaged over Germany are displayed in Fig. 9a. TQC

increases significantly with increasing CCN and shape parameters of the CDSD. For example, the medians of the different microphysics sub-ensembles vary by more than 400%, with TQC amounting to $0.01\,\mathrm{kg\,m^{-2}}$ in sub-ensemble nu0m and $0.044\,\mathrm{kg\,m^{-2}}$ in sub-ensemble nu8p. The comparison of sub-ensembles sharing identical CDSD shape parameters shows an increase of TQC by up to 300% when increasing CCN concentrations from maritime to polluted conditions (compare sub-ensembles nu0m and nu0p in Fig. 9a). Similarly, the change from the broadest to the narrowest CDSD enhances TQC by

roughly 150%. These values are much larger compared to the impact of microphysical perturbations on precipitation. An important implication from Fig. 9a is that IBC perturbations cannot cover the variability due to microphysical uncertainties on



cloud forecasts, which manifests by marginal or no overlap of the distributions which have different CCN and CDSD configurations.

The forecasted cloud fractions also systematically increase with an increase of CCN and shape parameters (Fig. 9b), in agreement with TQC. Compared to the pristine sky sub-ensemble (nu0m), medians of the numbers of cloudy grid points [(TQC) $> 50\,\mathrm{g\,m^{-2}}$] are increased by 35% in nu8m and 91% in nu8p simulation. Compared to TQC, a change of CDSD shape parameters shows only minor differences of cloud fraction in continental and polluted CCN conditions (e.g. nu8c and nu8p in Fig. 9b), presumably due to the atmospheric condition like humidity, which gives upper bounds of total cloud cover. Hence variability of CCN concentrations and CDSD shapes becomes less important and IBC uncertainty, which predominantly triggers convection and determines the upper bound of cloud coverage, governs the variability of spatial cloud distributions.

Vertically integrated rain water content (TQR) averaged over Germany shows a systematic but opposite response compared to TQC (Fig. 9c). TQR decreases with increasing CCN and shape parameter of CDSD and adumbrates the systematic impact found for precipitation. Compared to TQC the variability caused by microphysical perturbations becomes smaller, for instance, the TQR medians of sub-ensemble nu0m amounts to $0.033\,\mathrm{kg\,m^{-2}}$, and nu8p to $0.014\,\mathrm{kg\,m^{-2}}$, indicating an increase from sub-ensemble nu8p to nu0m by roughly 240%.

The steady decreasing systematic impact of the microphysical perturbations on cloud content, rain water content and eventually precipitation hints towards some kind of buffering effects or compensating processes that reduce the large, positive impact on clouds and eventually even turn it into a negative impact with respect to the rain production. Recent works by Barthlott et al. (2022) and Baur et al. (2022) shed light on those processes. One major process is the reduction of warm rain processes. The suppression of collisional growth of cloud droplets in polluted CCN conditions leads to less production of rain components, and small droplets become more likely to evaporate. Moreover, cloud optical properties are influenced as well through changes of the droplet effective radius. That can affect the radiative energy supply that triggers succeeding convection.

### 4.4 Systematic assessment of the relative impact

Finally we attempt to put the findings on statistically more solid grounds and use 180-member ICON-D2-EPS experiments performed for five days in August 2020. The classification into distinct weather situations with different synoptic control on cloud and precipitation results in three weakly and two strongly forced days (see Table 1). The regime dependent relative impact of the various perturbations is computed as follows: first, the absolute difference of every individual member to its corresponding sub-ensemble mean is calculated, secondly, its relative difference is calculated based on its sub-ensemble mean, for every sub-ensemble and every day separately (as in Sect. 4.1 and shown in Fig. 4). Thirdly, the median, the interquartile range and the 5th and 95th percentiles are computed by aggregating the days for each synoptic forcing separately (i.e. 360 samples for strong forcing and 540 for weak forcing). Finally, the samples are bootstrapped 100 times with replacement to get statistically robust results, and the mean of the 100 medians, interquartile ranges and percentile values are finally depicted in Fig. 10. This procedure takes into account the different mean values of distinct sub-ensembles on different days (see Table 1) and guarantees a fair comparison.




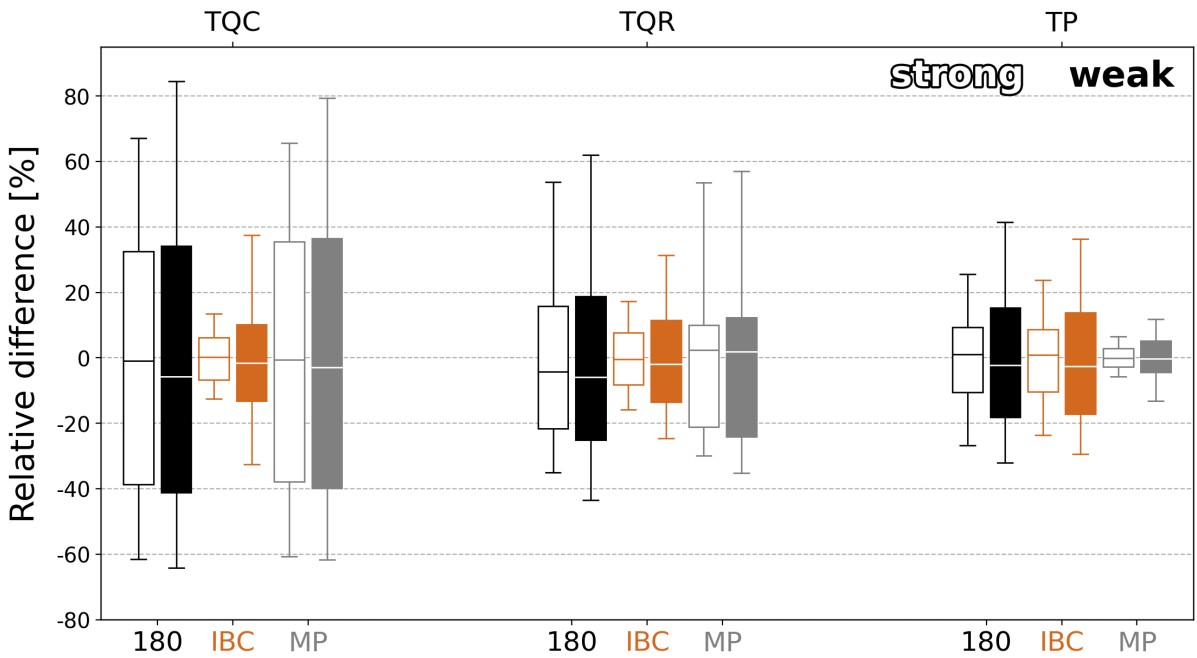

**Figure 10.** Relative differences of full 180 (black), averaged IBC (orange) and combined microphysical perturbations (grey) aggregated over five days in August 2020. Relative differences on total column cloud water content (TQC), total column rain water content (TQR) and total precipitation (TP) are displayed.

First, in the 180-member ensemble with IBC and combined microphysical uncertainties the 90% confidence interval (given by the 5th and 95th percentiles) of total precipitation of single experiments deviates from the ensemble mean by +41% to -32%, with an interquartile range between +15% to -18% during weak forcing. The impact of IBC perturbations on the 90% confidence interval shows a range of +36% to -29% in daily sums during weak forcing (orange boxes of IBC subensemble in Fig. 10). The variability is smaller and amounts to ±23% during strong forcing. The medians have a slightly negative bias for the weak forcing cases because its precipitation distribution is slightly positive-skewed, i. e. the mean is larger than the median. That might be an artefact of the given sample size.

Combined microphysical perturbations (grey bars in Fig. 10a) show a relative impact of +12% to -13% for the weak forcing cases, and ±6% for the strong forcing cases. Thus precipitation amounts are twice as sensitive to microphysical perturbations during weak control. While the interquartile range of the full 180-member ensemble and the individual IBC sub-ensembles is similar (between +16% and -18%), the 95th percentile of the 180-member ensemble representing the highest amounts of individual members surpasses those of the IBC sub-ensemble by 5% for weak forcing situations.

The same methodology is applied on convective clouds represented by the daily averaged vertically integrated cloud water content (TQC in Fig. 10). Several differences compared with the impact on precipitation become evident. First, the mean amount of TQC has a strong sensitivity to microphysical choices. For instance, the 180-member ensemble mean TQC is



$0.645\,\mathrm{kg\,m^{-2}}$, but $0.234\,\mathrm{kg\,m^{-2}}$ for the IBC sub-ensemble nu0m and $1.138\,\mathrm{kg\,m^{-2}}$ for that of nu8p on the weak forcing case 11 August. Similarly, TQCs for nu8p are 4 to 5 times as large as those for nu0m in the other cases (not shown). Moreover, the impact shows less dependence on synoptic control. microphysical perturbations show larger amplitudes on clouds than on precipitation, and their impact exceeds the impact of IBC uncertainty. The relative impact of microphysical perturbations on TQC ranges between +66% and -60% for strong forcing, and between +80% and -62% for weak forcing. Forecast variability

is increased by +47% when taking the microphysical uncertainties into account. The variability of CCN and CDSD plays a larger role in narrower CDSD or higher CCN conditions, similar to the impact on precipitation. Likewise the pure IBC impact on TQC is in line with that on precipitation, as the variability of TQC 90% confidence interval ranges +37% to -33% for the weak forcing cases, and +14% to -13% for the strong forcing cases.

     In the same way, impact on vertically integrated rain water content is also illustrated (TQR in Fig. 10). The impact on TQR

is systematic and lies between the impact on TQC and on precipitation. Microphysical perturbations show larger impact than IBC perturbations. The relative impact of microphysical perturbations on TQR ranges between +54% and -30% for strong forcing, and between +57% and -35% for weak forcing. Forecast variability is again increased by +31% when taking the microphysical uncertainties into account. The relative impact of IBC perturbations on TQR ranges between +17% and -16% for strong forcing, and between +31% and -25% for weak forcing.

Overall, microphysical uncertainty plays a more important role in the prediction of cloud and rain water content than IBC uncertainty, but the impact is buffered during warm rain processes. The buffering effect that counteracts to microphysical perturbations discussed in Sect. 4.3 is thus clearly quantified. The microphysical impact on the 95th percentile value amounts to +79% for TQC, 57% for TQR and 12% for TP. Conversely, the role of IBC uncertainty systematically increases from TQC, over TQR to precipitation. For instance, the interquartile range of the impact lies between +14% to -13% for TQC, +17% to

-16% for TQR and ±23% for TP during strong synoptic control.

## 5   Summary and concluding remarks

The relative importance of microphysical uncertainties on cloud and precipitation forecasts implemented in the operational ICON-D2-EPS is assessed on different spatial and temporal scales for five real cases in central Europe. The two-moment bulk microphysics scheme of Seifert and Beheng (2006) used in ICON-D2-EPS predicts next to the mass concentration of different

hydrometeors their number density and thus allows the calculation of the particle size distribution. In the present study we perturb two microphysical parameters that are poorly constrained by observations. Those constitute the cloud condensation nuclei (CCN) concentration, currently not considered in operational ensemble forecasting, and the shape parameter of the cloud drop size distribution (CDSD), currently kept constant. Their individual and combined relative impact is estimated in the presence of initial and boundary condition uncertainty (IBC) available from operational ensemble forecasting at Deutscher

Wetterdienst. Nine different set-ups of such combined microphysical perturbations run with 20 different IBC add up to a 180-member ensemble forecast. Additionally the relative impact is examined conditional to the prevailing weather situation classified with the convective adjustment time scale.





The close inspection of individual ICON-D2 experiments indicates a large variability due to IBC uncertainty in combination with the considerable variability due to microphysical uncertainties within the nine individual IBC sub-ensembles (Fig. 3). This

illustrates the necessity to be cautious when interpreting results based on a deterministic approach only to evaluate uncertainty. The use of a full ensemble modelling system including various key sources of uncertainty as done in this study is essential to assess their relative importance. This issue becomes even more relevant when inspecting smaller spatial and temporal scales.

Overall, combined microphysical uncertainties have a relevant impact on both amount and spatial variability of precipitation forecasts. The relative impact of pure microphysical perturbations is a third compared to the impact due to IBC perturbations

regarding spatially averaged precipitation totals over a domain as large as Germany, and affect the location of individual convective cells (O(10 km)). The impact of the combined microphysical perturbations on the spatial rainfall pattern is dominated by the CCN perturbations on average. The importance of the uncertainty is highly case dependent like other subgrid-scale parametrisation schemes such as the stochastic boundary-layer scheme (Hirt et al., 2019; Keil et al., 2019).

The impact of the different perturbations on precipitation can be quantified as follows:

– The impact on daily area-averaged precipitation (TP) depends on the synoptic control and is larger during weakly forced situations. The impact of pure IBC perturbations on the 90% confidence interval (given by the 5th and 95th percentile) of TP of single experiments ranges between +38% and -32% during weak forcing and ±25% during strong forcing.

    – Combined microphysical perturbations show a relative impact of +12% to -13% for the weak forcing cases, and ±6% for the strong forcing cases. Thus precipitation amounts are twice as sensitive to microphysical perturbations during weak

455       control.

    – CCN and CDSD perturbations show a large sensitivity to the other background (fixed) microphysics choice. That stems from the systematic behaviour of the responses to different microphysics conditions. Microphysical perturbations have systematic effects whereas IBC perturbations are likely to have stochastic effects.

    – While the interquartile range of TP of the full 180-member ensemble and the nine IBC sub-ensembles is similar, the 95th

460       percentiles of the 180-member ensemble surpass those of the IBC sub-ensemble by 5%. Thus, the combination of IBC and microphysical perturbations especially affects the magnitude of the extremes. The spatially and temporally averaged precipitation of extreme ensemble members exceeds the ensemble mean by 50%.

    – During weak control CCN and CDSD perturbations have a systematic impact on the intensity and location of individual convective cells identified in the present study with hourly rain rates, and its spatial variability amounts to O(10km)

465       quantified with FSS believable scales. In contrast, IBC perturbations scramble the precipitation pattern during weak control and result in twice the location uncertainty.

    – During weak control CCN perturbations cause a larger impact on spatial variability of precipitation forecasts than CDSD. Individual perturbations of CCN and CDSD have larger impacts when the other configuration is the narrower CDSD or polluted CCN condition, respectively.



470 Different from the impact on precipitation, the increase of CCN concentration and shape parameter of CDSD has a large positive impact on the production of cloud water content and forms horizontally larger clouds. The impact of combined microphysical perturbations on domain averaged TQC is not very sensitive on synoptic control and ranges between +65% and -60% in the strong forcing condition, and between +79% and -62% in the weak forcing. The impact on TQR also shows larger sensitivity to microphysical perturbations than to IBC uncertainty with the range of relative impact between +54% and -30%

475 for strong forcing, and between +57% and -35% for weak forcing. Thus the considerable impact on cloud variables does not directly translate into precipitation amounts. This implies that some microphysical processes or feedbacks are compensating for the impact. The systematic behaviour of cloud variables is consistent with previous studies (Seifert et al., 2012; Igel and van den Heever, 2017a; Wellmann et al., 2020; Zhang et al., 2021), and further discussion about the detailed processes seen from the deterministic perspective can be found in Barthlott et al. (2022) and Baur et al. (2022). Not surprisingly, IBC uncer-

480 tainty contributes less to TQC and TQR than microphysical uncertainty, especially in strong synoptic-forcing situations when cloud variables are less sensitive to IBC perturbations.

 Our results suggest that the consideration of CCN and CDSD uncertainties increases precipitation variability and can contribute to the reduction of the long-standing issue of underdispersion of near surface variables in convective scale EPS forecasts (see references in e.g., Keil et al., 2019) and thus ultimately benefit the improvement of NWP ensemble forecasting. It is be-

485 yond this study to assess to what extent the microphysical perturbations contribute to a better probabilistic forecasting skill compared to observation. Given the increasing importance of satellite observations used in convective scale data assimilation the systematic impact of microphysical uncertainties will attract interest in future. Microphysical uncertainties strongly influence forecasts of cloud coverage and droplet sizes, both representing important ingredients used in satellite forward operators to compute synthetic reflectances (e.g. Scheck et al., 2020) to be used in data assimilation algorithms.

490 *Code and data availability.* The ICON codes and data of the initial and lateral boundary conditions are available upon request with permission from the Deutscher Wetterdienst (DWD).

*Author contributions.* CK and CB oversee the project. CB and AZ designed the microphysical perturbations and TM set up the numerical model and carried out the experiments. TM prepared the manuscript with contributions from all co-authors. CK internally revised the manuscript and supervised the whole work.

495 *Competing interests.* The authors declare that they have no conflict of interest.



*Acknowledgements.* This research has been performed within projects B3 of the Transregional Collaborative Research Center SFB/TRR 165 "Waves to Weather" funded by the German Research Foundation (DFG). The authors wish to thank the Deutscher Wetterdienst (DWD) for providing the ICON model code and analyses datasets and Robert Redl and Fabian Jakub (LMU) for technical help.





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
