# Peer review of "The impact of microphysical uncertainty conditional on initial and boundary condition uncertainty during different synoptic control"

_Weather and Climate Dynamics, 2022_

## Referee Comment (RC1)

**Review on "Impact of combined microphysical uncertainties on convective clouds and precipitation in ICON-D2-EPS forecasts using different synoptic controls" by T. Matsunobu *et al.***

This paper examines the contributions of microphysics uncertainties arising from two sources, the cloud condensensation nuclei (CCN) concentration and the shape of the parameter of the cloud drop size distribution (CDSD). These impacts are investigated in 5 cases of precipitation over Germany in August 2020, including 2 strong forcing and 3 weak forcing situations. The work is performed using the convective-scale ICON-D2-EPS with 180 members. The contributions of microphysical uncertainties and initial and boundary condition uncertainties (IBC) are primarily examined on precipitation forecasts, with a focus on intensity and spatial variability, and secondly on cloud and rain water contents. While the impact of IBC overall dominates for precipitation forecasts, cloud and rain water contents appear to be more sensitive to the microphysical uncertainties.

The paper is well-written and provides a number of new and interesting results. It will surely contribute to a better understanding of model uncertainties, with potential applications for the design of EPS. In addition, the methodology is sound and the in-depth analysis with several diagnostics is valuable.

My main comments, detailed below, would recommend a simplification or clarification of some results, and a potential re-organization of the manuscript plan. For that purpose, I consider major revisions are required before publication in Weather and Climate Dynamics.

**Major comments**

• 1. My main concern is about the huge volume of information on some figures, in particular I found the readibility/understanding of Figure 3 particularly challenging (especially the top panel). Would it be possible to think about a new design that would make the interpretation easier ?

• 2. Different ensembles are examined in the paper (IBC, MP, CCN, CDSD). It would be very helpful to clearly define these sub-ensembles in a table (rather than in the text), with information on the number and size of the associated sub-ensembles.

• 3. As microphysics perturbations more directly impact cloud and rain water contents than precipitation, I think it would be more natural to start section 4 with the results of 4.3. Such a re-organization would imply a non-negligeable work but the resulting manuscript should be more consistent.

• 4. The study is based on the in-depth analysis of 2 cases, and more statistically robust results are computed with 5 situations. This is a small sample to draw conclusions, however I understand that running additional cases is beyond the scope of the paper for computational reasons. At least I seems important to underline this limitation in the conclusions.

**Specific comments**

• 1. L37-39 "The impact of parameter perturbations ... using a variety of NWP models and schemes" : add references to studies.
• 2. Figure 6 : add Time [utc] as x-axis legend.
• 3. Section 4.2 : add the definition of FSS somewhere.
• 4. L267-268 : do you think the differences of sampling size for the different ensembles can impact the results ?
• 5. L342-344 : is it only a spin-up and/or the nature of precipitation that explain the differences between nighttime and daytime rainfall ?
• 6. Figure 9 legend : (c) domain-averaged total column rain water content.

• 7. Figure 10 : It would be interesting, for each variable, to discuss the statistical significance of differences observed between the 3 sub-ensembles, and between weak and strong forcing situations.

---

## Author Comment (AC1)

**Response to Reviewer 1 on the Manuscript wcd-2022-17 entitled "Impact of combined microphysical uncertainties on convective clouds and precipitation in ICON-D2-EPS forecasts during different synoptic control"**

by Takumi Matsunobu et al.

**1 General comment**

This paper examines the contributions of microphysics uncertainties arising from two sources, the cloud condensensation nuclei (CCN) concentration and the shape of the parameter of the cloud drop size distribution (CDSD). These impacts are investigated in 5 cases of precipitation over Germany in Au- gust 2020, including 2 strong forcing and 3 weak forcing situations. The work is performed using the convective-scale ICON-D2-EPS with 180 members. The contributions of microphysical uncertainties and initial and boundary condition uncertainties (IBC) are primarily examined on precipitation fore- casts, with a focus on intensity and spatial variability, and secondly on cloud and rain water contents. While the impact of IBC overall dominates for precipitation forecasts, cloud and rain water contents appear to be more sensitive to the microphysical uncertainties. The paper is well-written and provides a number of new and interesting results. It will surely contribute to a better understanding of model uncertainties, with potential applications for the design of EPS. In addition, the methodology is sound and the in-depth analysis with several diagnostics is valuable. My main comments, detailed below, would recommend a simplification or clarification of some results, and a potential re-organization of the manuscript plan. For that purpose, I consider major revisions are required before publication in Weather and Climate Dynamics.

We would like to thank the reviewers for their constructive comments, which will help to improve further the quality of the manuscript. Note that we changed the title to 'The impact of microphysical uncertainty conditional on initial and boundary condition uncertainty during different synoptic control' and the co-author list due to major modifications and considerable re- writing. The answers to the reviewer's remarks describing the corrections made in the manuscript, are written in blue hereafter.

**2 Major comments**

– 1. My main concern is about the huge volume of information on some figures, in particular I found the readibil- ity/understanding of Figure 3 particularly challenging (especially the top panel). Would it be possible to think about a new design that would make the interpretation easier ?

We re-designed Figure 3 and reformulated the explanation in Section 4.1.

- 2. Different ensembles are examined in the paper (IBC, MP, CCN, CDSD). It would be very helpful to clearly define these sub-ensembles in a table (rather than in the text), with information on the number and size of the associated sub-ensembles.

Thank you for pointing that out. We changed the text accordingly at several places (at the beginning of the Experimental desgin and the results sections) to clearly describe our subsampling strategy used to quantify the relative impact of different uncertainties by analysing different sub-ensembles.

- 3. As microphysics perturbations more directly impact cloud and rain water contents than precipitation, I think it would be more natural to start section 4 with the results of 4.3. Such a re-organization would imply a non-negligible work but the resulting manuscript should be more consistent.

We agree, a process chain like structure of the results would be a sensible approach. However, we put the weather regime dependence of precipitation at the heart of our investigation and start our line of argument very broad and detailed with an illustration of the precipitation difference of all 180 ICON-D2 ensemble members for two different cases. Subsequently, we compress the data and show boxplots to focus on the relative contribution of the various uncertainties. Having found the largest impact during weak synoptic control, we then examine the spatial predictability of precipitation. After that we turn to cloud and rain water contents. We changed the order in Section 4.3 going 'backwards the process chain' from precipitation to cloud water content so that readers can more easily follow the argument from precipitation at the ground to microphysical impacts within clouds.

- 4. The study is based on the in-depth analysis of 2 cases, and more statistically robust results are computed with 5 situations. This is a small sample to draw conclusions, however I understand that running additional cases is beyond the scope of the paper for computational reasons. At least I seems important to underline this limitation in the conclusions.

We understand your concern on the limited data base. We added a sentence in the conclusions mentioning the limitation of this study as follows:

> We caution the limited dataset covering five days in August 2020 only. More robust results require a larger data base containing more cases that comprise different synoptic conditions. Based on the five cases we cannot draw general conclusions.

**3 Specific comments**

- 1. L37-39 "The impact of parameter perturbations ... using a variety of NWP models and schemes" : add references to studies

We agree and added the following text:

The impact of parameter perturbations in microphysical parametrisations has been studied extensively in mostly single deterministic idealized (e.g. Grant and van den Heever, 2015; Glassmeier and Lohmann, 2018; Heikenfeld et al., 2019; Chua and Ming, 2020; Wellmann et al., 2020) or realistic (e.g. Bryan and Morrison, 2012; Barthlott and Hoose, 2018; Schneider et al., 2019; Baur et al., 2022) simulations using a variety of NWP models and schemes.

– 2. Figure 6 : add Time [utc] as x-axis legend.

Added.

– 3. Section 4.2 : add the definition of FSS somewhere.

We added the definition of the FSS in Sect. 4.2 with its mathematical interpretation.

The definition of the FSS is given by

$$FSS = 1 - \frac{\sum (f_A - f_B)^2}{\sum f_A^2 + \sum f_B^2} \tag{1}$$

where $f_A$ and $f_B$ represent the fraction of rainy grid points in fields $A$ and $B$, respectively, at which the precipitation amount is exceeding a certain threshold value. The second term on the right hand side represents the ratio of the mean squared error (MSE) of the fraction fields $A$ and $B$ to their maximum possible MSE.

– 4. L267-268 : do you think the differences of sampling size for the different ensembles can impact the results ?

Thank you for pointing this out. To check the sample size effect, we computed the FSS with different sample sizes (Fig. R1.1, as in Fig. 6 in the manuscript). The impact of different sample sizes is very small for sample sizes used in this study (n=1710, 720, 180, top row in Fig. R1.1), only for two-digit (and smaller) sample sizes there are slight modifications of the FSS and believable scales. Therefore we conclude that the differences caused by the sample size are negligible in this study.

Note that there was a typo on the sample size of combined microphysical perturbations in the manuscript. 720 is the correct value (was falsely 7200). This is corrected.

– 5. L342-344 : is it only a spin-up and/or the nature of precipitation that explain the differences between nighttime and daytime rainfall ?

We believe that the differences are mainly caused by spin-up effects. A similar discussion was raised in Barthlott et al. (2022, ACP), in which the impact of CCN and CDSD perturbations on precipitation was addressed and a smaller impact of microphysics perturbations at short lead times was found. They write that "The comparably small spread in precipitation intensities for both the CCN and shape parameter runs during the nighttime precipitation maximum on 2 June 2016 could be explained by the fact that this maximum occurs during the first hours of the simulation. In that time, the spin-up effects and the adjustment to the driving coarser-scale model are still in effect, which could dampen

[Figure]

**Figure R1.1.** Time series of FSS values of hourly precipitation calculated on scales ranging from 2 to 560 km across the German domain for the weak forcing case 11 August. Each panel shows the mean of re-sampled data with different sample sizes (from 1710 to 3). The black lines show believable scales of mean FSS. The red lines (right axis) show the time series of the entire IBC sub-ensemble mean 99th percentile value of hourly precipitation. All valid for of the IBC sub-ensemble.

the impacts of the microphysical uncertainties assessed here. A similar, smaller impact of microphysics perturbations at short lead times was found in further sensitivity experiments for 11 June 2019, initializing the model at 18:00 UTC (not shown)." We added a citation to our manuscript.

In addition we compared hourly precipitation maps showing nighttime and daytime precipitation to examine if the nature of precipitation is completely different (Fig. R1.2). Both nighttime and daytime precipitation fields show clear rain bands across Germany with intermittent strong precipitation. Both snapshots represent convective precipitation events that are driven by synoptic-scale flow thus sharing a similar nature of precipitation.

We therefore conclude that the small impact on nighttime precipitation is largely due to spin-up effects. The similar precipitation characteristics of nighttime and daytime precipitation hint towards the same nature of precipitation.

– 6. Figure 9 legend : (c) domain-averaged total column rain water content.

[Figure]

**Figure R1.2.** Two snapshots of hourly precipitation rates valid at 02 UTC (nighttime precipitation, left) and at 15 UTC (daytime precipitation, right). Data of member 1 of a nu0c ICON-D2 simulation on 17 August 2020 (strong-forcing case) is shown.

Corrected.

– 7. Figure 10 : It would be interesting, for each variable, to discuss the statistical significance of differences observed between the 3 sub-ensembles, and between weak and strong forcing situations

We agree. However, as mentioned above in major comment 4, robust results including a statistical significance test necessitate a larger data base that is beyond the scope of the present study but is planned in future work.

---

## Author Comment (AC2)

**Response to Reviewer 2 on the Manuscript wcd-2022-17 entitled "Impact of combined microphysical uncertainties on convective clouds and precipitation in ICON-D2-EPS forecasts during different synoptic control"**

by Takumi Matsunobu et al.

**1 General comment**

The paper "Impact of combined microphysical uncertainties on convective clouds and precipitation in ICON-D2-EPS forecasts during different synoptic control" presents the impact of the perturbation of two parameters of the microphysics scheme, individually and combined, with respect to perturbation of initial and boundary conditions, in a convection- permitting ensemble based on the ICON model. The topic is an interesting one and the method of investigation proposed by the authors is sound and insightful. However, I have found the paper very difficult to read, partly due to the language and party due to the way the results are presented and discussed. The reader is not well guided through the interpretation of the results and some statements and conclusions not well justified. Since the material of the research is very good and the methodology for sure interesting for the scientific community, I suggest to the authors to carry out a major re-writing of their analysis and findings. I provided a list of points where improvements would be beneficial. I therefore recommend the paper for a major revision.

We would like to thank the reviewers for their constructive comments, which will help to improve further the quality of the manuscript. Note that we changed the title to 'The impact of microphysical uncertainty conditional on initial and boundary condition uncertainty during different synoptic control' and the co-author list due to major modifications and considerable re-writing. The answers to the reviewer's remarks describing the corrections made in the manuscript, are written in blue hereafter.

**2 Detailed comments**

– Page 1, line 5: inspect -> inspected. The abstract is a bit dense of data, it may be difficult for the reader to follow and get an informative summary about the paper content.

Corrected. We also reduced specific numbers in the abstract and modified the text for a better guidance to our main conclusion.

– Page 2, line 22-23. "The uncertainties originate from the chaotic nature of the atmospheric flow ..." This statement should be modified. The uncertainties originate from the unknown initial state and the imperfect representation of physical

processes, as well as from approximations in the model, but not from the chaotic nature of the atmosphere. The latter is instead the responsible of the large effect that the (even small) uncertainties have on the weather forecast.

Reply to the comments about Page 2 line 22-34 is put together. Please see reply below.

– Page 2 line 27: the chaotic nature of convection: I guess the authors mean the stochastic nature of convection? Chaoticity applies to the atmosphere as a whole, therefore the sentence would not have the intended meaning.

Please see reply below.

– Page 2 line 29-30: "An ensemble facilitates the distinction between systematic effects of perturbations and the chaotic signal." I think that this sentence needs to be better explained, for example, how does the ensemble facilitate this distinction?

Please see reply below.

– Page 2, line 31: represents -> is?

Please see reply below.

– Page 2, lines 31-32: data assimilation systems are for initial conditions only, not for IBC.

Please see reply below.

– Page 2 line 34: constitutes -> is constituted by? (or better: is)

We completely re-wrote the first half of the introduction and took your concerns into account.

– Page 2, line 35: "and to represent the subgrid-scale variability." Is this a reference to the intrinsically stochastic physics schemes?

Implicitly, yes. Nevertheless, the perturbations and scheme in this study does not consider the subgrid-scale variability. We added one paragraph in Sect. 2.2 mentioning that point.

The parameters describing the CCN concentration and the shape of CDSD are kept temporally and spatially constant throughout the simulation. Therefore the microphysical perturbations used in this study only represent model error due to the incomplete description of physical processes and do not include the subgrid-scale variability.

– Page 3 line 72: constitutes -> is constituted by?

We think 'constitutes' correctly represents our intention here because the shape of CDSD is one of the potential sources of microphysical uncertainty. It is not formed by microphysical uncertainty.

– Page 3, lines 80-81: please revise the language of this sentence

Rephrased the sentence like below:

The goal of the present study is to estimate the relative importance of microphysical uncertainties on precipitation in the presence of IBC uncertainties conditional to different synoptic control across central Europe. The microphysical perturbations comprise three different aerosol concentrations and three different shape parameters governing the cloud droplet size distribution (CDSD).

– Section 2.1. I find a bit confusing the model description. ICON-D2 is the operational 2 km run at DWD, and not the model name, while you use here the ICON model, in its limited-area version, at 2km, model version 2.6.2.2.

Please see reply below.

– Page 4, line 99: ICON-D2-EPS is not introduced, neither it is mentioned in Section 2.1 that an ensemble system is used. How is it built? I would suggest also here to refer to ICON-D2- EPS if the operational ensemble of DWD in mentioned, to an ensemble based on ICON at 2km if own experiments are made.

Considering two comments above, we agree that we were a little bit sloppy when referring to ICON-D2-EPS. We reformulated the paragraph, introduced ICON-D2-EPS with a reference to its official documentation and use the term 'ICON-D2 ensemble' for our 180-member simulations in the text, since the model domain is identical. We removed the phrase in the title and re-formulated the title accordingly.

The numerical simulations are performed with the ICON (ICOsahedral Non-hydrostatic, version 2.6.2.2) model in its limited-area mode ICON-D2 covering central Europe (see Fig. 2). The ICON-D2-EPS (Reinert et al., 2021) is the operational ensemble NWP system at Deutscher Wetterdienst (DWD) since February 2021. We use an almost equivalent configuration with few exceptions described below. ICON-D2 employs an icosahedral-triangular Arakawa-C grid with a grid spacing of 2 km (542040 grid points) and 65 vertically discretised layers from the ground to 22 km above mean sea level. Its dynamical core is based on the non-hydrostatic equations for fully compressible fluids as governing equations (see Zängl et al. (2015) for the details). Different from the operational configuration, the two-moment bulk microphysics scheme (Seifert and Beheng, 2006) is used to investigate the impact of number densities and the size distributions of cloud water droplets (by perturbing the CCN concentration and shape of CDSD, respectively, as in Barthlott et al., 2022). Note that the operationally used parameter perturbations in ICON-D2-EPS are turned off here to purely focus on the microphysical perturbations representing the model error.

– Page 6, line 143: I guess that something is missing in "is less about 2 hours"

To clarify the message, we modified the sentence like below.

The daily maximum $\tau_c$ is less than 2 hours on 17 August 2020 (red line in Fig. 2c and Table 1)

– Page 6, line 145: texture -> pattern? 'Pattern' fits better. Thank you.

– Page 7, line 148: boundary -> boundary layer? Corrected.

– Page 7, line 159: spanned by diverse IBC? Changed.

– Section 4: the first paragraph (page 7, lines 157-164) is very difficult to follow, could the authors explain better what they did? It is difficult to evaluate the results, since the method is not clearly described.

Following the reviewer's suggestion, we start Section 4 with explaining the subsampling approach in more detail. Thank you.

– Page 7, line 166: prototype? Have prototype cases been introduced?

'Prototype cases' indicate two of the five cases used for introducing the typical results for different synoptic forcing cases. It was rephrased along the change in discussion about Figure 3.

– Figure 3: I suggest to describe the meaning of the plot, first, since it is not a standard visualisation method (though appropriate for this paper and insightful, once it is explained): what is the meaning of the dots being in the upper or lower part, what is the meaning of being grouped together or sparse, in order to guide the reader to an understanding of the results. In the text some conclusions are drawn from inspecting the plots but the reader cannot understand on what they are based. An example: "At first sight and independent of the prevailing weather situation, the IBC perturbations largely control the precipitation amount." From what is this visible? I also suggest to revise the caption of the figure. In it is also mentioned "the nine coloured dots" but the coloured dots in the plot are many more than nine ...

Following the suggestion from the other reviewer, we redesigned Figure 3 to easily show and explain the highlighted points using the plot. The explanation in the main text was also reformulated.

– Page 9, line 180: where are the +3,5% and – 3,5% values visible? I think that in presenting this result it should be made the difference between the two extremes (polluted negative difference (less rain?), maritime positive difference)

We compared the mean values of blueish lines and reddish lines, although no line shows the mean values in the plot to avoid many overlapped lines. Another reason why we used the mean was that we wanted to compare the pure impact of individual CCN and CDSD perturbations without additional plots only for showing those impact. We hope that our point becomes understandable with the reformulated text.

– Page 9, line 185-186: "Shape parameters of CDSD also exhibit a systematic impact in the weak forcing situation": from what is this visible? The dots with different nuances of the same colour are more disperse, and systematically? Please mention this explicitly to help the reader.

In the revised paragraph, we explicitly mentioned the lines we want to compare.

– Page 9, line 191: can you explain better what is meant with reverse, with respect to which order?

We meant the exchange of the positions in the colour order between two dots next to, against the order of colours for the mean impact (lines). It does not mean the entire reversal of the colour order of the mean. To avoid confusing, we reformulated the text to indicate the specific example.

– Page 9, lines 192-195: I do not understand the meaning of this sentence, could this be clarified?

Considering the two comments above we reformulated the paragraph as follows:

> Interestingly, a closer inspection reveals that different IBC can completely reshuffle the rank of the individual members in a specific MP sub-ensemble. For instance, experiments with modest aerosol content but different shape of the CDSD show extremes for member 11 during weak control ( *nu8c* (dark green) shows the largest negative and *nu2c* (medium green) shows the largest positive impact, Fig. 3a) This non-systematic and highly varying response of precipitation to perturbed microphysical parameters of individual ICON-D2 experiments points towards a strong sensitivity to IBC. This finding illustrates the necessity to be cautious when interpreting results based on a deterministic approach only to evaluate uncertainty.

– Page 9, lines 197-200: I would suggest to explain more clearly what is shown in Figure 4, it is very difficult to follow (e.g. does "a sub-ensemble mean sharing the same unperturbed parameters" mean "the mean of the sub-ensemble sharing the same unperturbed parameters?")

Thank you for pointing that out. We changed the text accordingly at several places (at the beginning of the Experimental desgin and the results sections) to clearly describe our subsampling strategy used to quantify the relative impact of different uncertainties by analysing different sub-ensembles.

> To assess the relative contributions of the various uncertainties we extract different sub-ensembles from the large 180-member ensemble. First we focus on 9-member MP sub-ensembles in which each of the sub-ensemble members has different combinations of CCN and CDSD parameters but identical IBC to examine the relative contribution of the combined microphysical (MP) perturbations on precipitation. Since there are 20 IBC in the entire ensemble, there are 20 different MP sub-ensembles with nine members each. Likewise, there are nine 20-member IBC sub-ensembles, with one fixed combination of MP perturbations but 20 different IBC. This different subsampling perspective allows to draw conclusions on the relative impact of IBC uncertainty. Lastly, there are three 60-member CCN and CDSD sub-ensembles, respectively, that inform on their individual contribution.

– Page 10, line 213: to +22% to -20% ?

We corrected the language.

> increases to a range of +22% and -20%

– Page 10, line 219: "the interquartile range becomes smaller than those...": which interquartile range? I guess the one of the MP sub-ensemble, gray bars? Please specify. Does this also have a meaning? Perturbing the 2 microphysical parameters together (gray bars) leads to larger extremes than perturbing only one of the two parameters, but to smaller interquartile range than perturbing CCN with $\nu$ 2 and 8, as said. Can this also be explained?

We moved the sentence to the paragraph before and explained the implication given by the smaller interquartile ranges of MP than those of CCN perturbations.

Precipitation reacts more sensitive to microphysical perturbations during weak synoptic control. In this situation the interquartile range of the combined MP sub-ensemble (grey box) becomes smaller than those of the CCN sub-ensembles with fixed shape parameters (cyan boxes for fixed $\nu = 2$ and 8) corresponding to a narrower CDSD. Thus adding CDSD perturbations to CNN uncertainty renders the probability density function of the relative impact sharper as well as leads to an extension of the tails of the distribution (grey dots of MP sub-ensemble).

– Page 11, line 234-235: please substitute from "three of them" with "the experiments in panels b, c and d share with the experiment in panel a identical IBC, CCN concentration and shape parameter of CDSD, respectively".

We modified the sentence following the suggestion.

In Fig. 5 the transient character of individual cells is juxtaposed for four different experiments: three of them share the identical IBC (panels a, b and c), CCN concentration (panels a, b and d) and shape parameter of CDSD (panels a, c and d), respectively.

– Page 11, line 240: I am not sure if Luxemburg can be identified clearly on this map by a non- European reader, I suggest to change the reference (using lat lon lines? A red circle?)

Following the reviewers suggestion, we added a red circle in the plot and clearly specify the rain cells that we are discussing.

– Page 11, lines 242-250: I do not understand from what can be evaluated the effect of the CCN and CDSD perturbations from the plots of Figure 5. Comparing panels a and b one can see the impact of changing the shape parameter in the polluted setup, while comparing a and c one can see the impact of changing the CCN condition with shape parameter equal to 8. Therefore, from where can be evaluated that "The relatively small impact of CDSD perturbations in maritime CCN conditions", when there is only one plot referring to maritime conditions? And where is it shown that "Positions of strong rain cells are shifted by the CCN perturbation at a scale of 20-30 kilometres, whereas an increase of the shape parameter of CDSD hardly shows a clear difference."? Plots b and c look anyway very similar. Given to this, the conclusion drawn in lines 247-250 cannot be judged.

Thank you for pointing out this point. We withdrew the statement "Positions of strong rain cells..." from the visual inspections not shown in this paper. We agree that it is a strong conclusion stating without showing any plots for a maritime CCN conditions. Since the same conclusion is drawn from the following FSS analysis, we removed this statement about maritime CCN simulations from this paragraph.

– Page 13, line 260: why should the choice of the 99th percentile of precipitation as threshold guarantee that the number of grid points used for FSS calculation is constant?

Since the percentile threshold value is calculated using all the grid points in the evaluation domain, the number of grid points over the threshold always must be fixed unless the evaluation domain is changed. In this study, the number of grid points in the evaluation domain is 342*295 = 100890. Therefore the number of grid points above the threshold, regarded as a precipitated grid, was always 1008.

– Page 14, lines 275-279: I suggest to move this specification after Figure 8 has been introduced.

We moved this note toward the end of Section 4.2, after introducing all the believable scale plots.

– Page 17, lines 340-344. From panels 8c and 8d it appears that the impact of the CDSD perturbation starts to appear only at 7 UTC, while for CCN it only starts at 6 UTC. Since, as pointed out by the authors, there is continuous rainfall, this characteristic is a bit puzzling me. I understand the argument that microphysical perturbations may need a longer spin-up time to modulate the fields, but it seems to me a bit too regular, and the same for all the 3 experiments in each case. On top, the gray thin lines, for all the possible combinations of members, also start abruptly at, likely, 5 and 4 UTC respectively. I would suggest to the authors to check what happens here, since I am not sure that this can be explained by the physical spin-up of the perturbations.

To the first question about a spin-up effect was discussed in previous work by Barthlott et al. (2022, ACP), in which the impact of CCN and CDSD perturbations on precipitation was addressed and smaller impact of microphysics perturbations at short lead times was found. Quoting the sentences explaining this point from their paper, it is explained by a spin-up effect:

"The comparably small spread in precipitation intensities for both the CCN and shape parameter runs during the nighttime precipitation maximum on 2 June 2016 could be explained by the fact that this maximum occurs during the first hours of the simulation. In that time, the spin-up effects and the adjustment to the driving coarser-scale model are still in effect, which could dampen the impacts of the microphysical uncertainties assessed here. A similar, smaller impact of microphysics perturbations at short lead times was found in further sensitivity experiments for 11 June 2019, initializing the model at 18:00 UTC (not shown)."

We also show the time evolution of FSS on the other nighttime precipitation case, 18 August 2020 (Fig. R2.1). Although its nighttime precipitation is weaker than that on 17 August, it still can show a slow start of the believable scale evolution

[Figure]

**Figure R2.1.** MP sub-ensemble mean FSS values of hourly precipitation calculated across scales ranging from 2 to 560 km across the German domain for 18 August. The black lines show believable scales of mean FSS. The red lines (right axis) show the time series of mean 99th percentile value of hourly precipitation.

from 8 UTC. Therefore we would like to keep an emphasis of the explanation on a spin-up effect rather than suggesting another new hypothesis. Nevertheless, we agree that the number of case studies are still too small to generalise this conclusion.

To the second question about the abrupt increase of believable scale at 5 UTC in the weak-forcing condition, we answer that it is due to an occasional suppression of convective initiation in one ensemble member. In the member 10 of nu0c and nu2m simulations convective systems found in the same member of nu0m simulations hardly evolve and different convection occurs far from those in the nu0m simulation (Fig. R2.2). That caused by chance the abrupt increase of the believable scale in Fig. 8a and b. This is partly owing to the change in microphysical parameters, but largely owing to a couple of convective systems strongly sensitive to a small perturbations. This kind of sensitive convection is also an interesting topic, however we would like to avoid detailed discussion because it is off track from our main topic in this paper.

– Figure 9. Caption: in c is "total column rain water content". The unity of measure for cloud fraction is missing.

Since cloud fraction is a non-dimensional index, a blank should suit in this case.

[Figure]

**Figure R2.2.** As for Fig. 5, but hourly precipitation rate at 5 UTC of (left) nu0m and (right) nu0c simulation for 11 August 2020. Forecast member 10 over the entire Germany domain is shown.

– Section 4.3. When the plots are described, the experiments are mentioned with their label (e.g. nu8c) but there are no labels in these plots. I would therefore suggest to mention the colours, instead, or add the labels. Actually this problem has presented itself in other points through the paper.

I agree the reviewer's comment. Considering that we don't use the colour scheme in some plots of spatial variability, we would like to keep both the colour and experiment names throughout the text and have labels in legends of the plots.

– Page 18, lines 360-361: The meaning of this sentence is not clear.

We reformulated the sentences like below:

The forecast cloud fraction also systematically increases with higher CCN and shape parameters (Fig.9b), in agreement with the increase in TQC. Cloudy grid points are defined as a grid cells where TQC > 50 $\text{g m}^{-2}$. The medians of the cloud fraction in IBC sub-ensemble nu0m (light blue), nu8m (dark blue), nu0p (light red) and nu8p (dark red) are 0.29, 0.39, 0.47 and 0.55, respectively. Thus, cloud fraction increases with higher CCN or/and CDSD parameters by 35%, 62% and 91% relative to experiment nu0m. Compared to TQC, a change of CDSD shape parameters shows an only minor effect on cloud fraction in continental and polluted CCN conditions (e.g. nu8c and nu8p in Fig. 9b). This is presumably caused by ambient atmospheric conditions as, e.g., humidity sets an upper bound for total cloud cover.

– Page 19. Since the description of Figure 10 starts from the bars on the right, I suggest to signalise this to the reader, e.g. at line 391: "total precipitation, TP (right column of Figure 10)". In the text is also mentioned a Figure 10a, not present.

Considering your and the other reviewer's comment, we swapped the position of the boxplots for TQC and TP. Following that the order of description of TQC and TQR in Fig. 10 was also swapped to naturally guide readers from left to right.

– Page 19-20, lines 404-405: these figures are not visible form the plot, please remove the sentence.

Removed the sentence.

– Page 17, line 355. "These values are much larger compared to the impact of microphysical perturbation.": We agree with the authors that the effect on TQC is remarkable, in particular when the variability of the IBC sub-ensemble values is compared with the variability between different setup of the microphysics parameters. However, it is difficult to compare even the relative increase or decrease of TQC with the one of precipitation, since the two variables and not directly comparable, in terms of unity of measure and scale of their distributions. I suggest to find a different formulation to express the result, maybe lines 356- 358 are already enough.

Please see reply below.

– Page 20, lines 410-411: this result cannot be discuss from Figure 10, please remove it.

Please see reply below.

– Page 20, lines 422-425: I would suggest to remove these lines, since these numbers are likely dependent on the considered cases and anyway very difficult to compare (see my comment before).

Those 3 comments above concern the comparability of relative impact on TP and TQC or TQR. We understand that TP is an accumulation over 24 hours and sedimentation on the ground while TQC and TQR are temporally averaged snapshot values vertically integrated over the entire column of the atmosphere. However, precipitation largely (around 95% in our simulations) comes from sedimentation of rain water and units of TP and TQR are the same ( $kg/m^2$ ), thus it can be imagined that an average impact on TQR can directly affect TP with the similar amplitude of impact (of course this is a wrong assumption). We think that we can quantify the contribution of so-called buffering effects (i.e. processes saturating impact on ground precipitation) by comparing the impact on TQR and TP. Therefore we would like to keep our conclusion as it is, but we added a notice pointing out this problem in the conclusion as follows.

Note that we compare rainfall accumulations at the ground with averages of 24 hourly snapshot scenes of vertically integrated cloud and rain water to facilitate a comparison of the respective contribution.

– Page 20, line 426: I do not think that the microphysical uncertainties have been in this work implemented in the operational ICON-D2-EPS, right?

You are right. The microphysical uncertainties are not operationally implemented. For a clearer statement, we modified the sentence and use the term 'ICON-D2 ensemble' instead.

– Page 21, lines 440: I do not understand the meaning of this sentence. Does it refer to the usefulness of a probabilistic approach at all? And how uncertainty would be quantified, in that case?

Thank you for pointing out this problem of language. We would like to state the necessity of including IBC perturbations to assess impact of other perturbations (e.g. microphysics) because the perturbations might be very sensitive to small

differences in IBC. I agree with the reviewer that the method presented in this paper is difficult to be applied for a quantification of 'uncertainty' itself. Following the comment, we modified the sentence as follows:

> Importantly, a close inspection of the synergistic impact of microphysical uncertainties in the presence of different IBC on precipitation indicates a strong sensitivity to IBC uncertainty (Fig. 3). This illustrates the necessity to be cautious when interpreting results based on a deterministic approach only to evaluate impact of uncertainty. The use of a full ensemble modelling framework including various key sources of uncertainty as done in this study is essential to assess their relative importance.

– Page 21-22, Summary and concluding remarks. There are some points in this section which I would ask the authors either to reconsider or to reformulate, but since the conclusions are mainly drawn from the results presented in section 4, which I suggest to consider for a major modification, I prefer not to go into details of Section 5 at present. I suggest to re-write Section 5 once the results are presented in a clearer way.

Summary and concluding remarks are reformulated considering the reviewer's comments and the updated discussion in the result section.

---

## Referee Report (RR1)

**Review on "The impact of microphysical uncertainty conditional on initial and boundary condition uncertainty during different synoptic control" by T. Matsunobu *et al.***

This revised manuscript has been significantly re-written and the authors properly answered my remarks. I still have some minor comments detailed below, after what I will consider the manuscript suitable for publication in Weather and Climate Dynamics.

**Specific comments**

• 1. L16 : Cloud and rain water contents.

• 2. L135-139 : this paragraph would be easier to understand if moved at the end of section 2.2 after the presentation of IBC and microphysics uncertainties.

• 3. L140 : The initial conditions of the IBC uncertainty : awkward, please reformulate.

• 4. L158 : emulates.

• 5. L182 : recall that $\tau_c$ is the convective adjustment time scale.

• 6. Legend of Table 1 - daily precipitation of IBC sub-ensemble mean of control : what do you mean by "mean of control" ?

• 7. L215-216 : there are three 60-member CCN and CDSD sub-ensembles. These sub-ensembles also take into account IBC uncertainty, hence it is not appropriate to call them CCN and CDSD. I would rather consider sixty 3-member CCN and CDSD sub-ensembles to properly evaluate the individual impact of microphysics perturbations. L380 you say that there are 180 combinations of ensemble members for CCN and CDSD sub-ensembles, which would mean that you considered sixty 3-member sub-ensembles instead of three 60-member sub-ensembles (in that case the number of combinations would be 5310). Can you clarify this point ?

• 8. L219-220 - the 24-hr accumulated area-averaged precipitation of all 180 ensemble members is shown in Fig 3 : please reformulate because precipitation differences are shown.

• 9. Legend of Fig.3 - coloured lines show average relative differences of them : awkward, please reformulate.

• 10. L245 : impacts.

• 11. L311 : CNN $\rightarrow$ CCN.

• 12. L536 : The impact of combined microphysical perturbations ... show a relative impact. Double use of "impact" is awkward, please reformulate.

• 13. L548-549 - Forecast variability is again increased by +31% when taking microphysical uncertainties into account : increased compared to what ? (same comment L555).

• 14. L569 : add these values are for the weak synoptic control.

• 15. L569-570 - the role of IBC uncertainty systematically increases from TQC, over TQR to precipitation : this is true only for strong synoptic control.

• 16. L604-605 : daily precipitation cannot take values between +38% and -32%, these values refer to relative differences, please reformulate.

• 17. At several places the term "90% confidence interval" is wrongly used. What you are looking to is not a confidence interval but the central 90 % inter-percentile range of the distribution of precipitation differences.